# Heterogeneous Clinical Phenotypes of dHMN Caused by Mutation in *HSPB1* Gene: A Case Series

**DOI:** 10.3390/biom12101382

**Published:** 2022-09-27

**Authors:** Xiya Shen, Jiawei Zhang, Feixia Zhan, Wotu Tian, Qingqing Jiang, Xinghua Luan, Xiaojie Zhang, Li Cao

**Affiliations:** Department of Neurology, Shanghai Jiao Tong University Affiliated Sixth People’s Hospital, 600 Yishan Road, Shanghai 200233, China

**Keywords:** HSPB1, dHMN, CMT2F, heat shock protein, restless legs syndrome

## Abstract

Mutations in *HSPB1* are known to cause Charcot-Marie-Tooth disease type 2F (CMT2F) and distal hereditary motor neuropathy (dHMN). In this study, we presented three patients with mutation in *HSPB1* who were diagnosed with dHMN. Proband 1 was a 14-year-old male with progressive bilateral lower limb weakness and walking difficulty for four years. Proband 2 was a 65-year-old male with chronic lower limb weakness and restless legs syndrome from the age of 51. Proband 3 was a 50-year-old female with progressive weakness, lower limbs atrophy from the age of 44. The nerve conduction studies (NCS) suggested axonal degeneration of the peripheral motor nerves and needle electromyography (EMG) revealed chronic neurogenic changes in probands. Open sural nerve biopsy for proband 2 and the mother of proband 1 showed mild to moderate loss of myelinated nerve fibers with some nerve fiber regeneration. A novel p.V97L in *HSPB1* was identified in proband 3, the other two variants (p.P182A and p.R127W) in *HSPB1* have been reported previously. The functional studies showed that expressing mutant p.V97L HSPB1 in SH-SY5Y cells displayed a decreased cell activity and increased apoptosis under stress condition. Our study expands the clinical phenotypic spectrum and etiological spectrum of *HSPB1* mutation.

## 1. Introduction

Mutant HSPB1 (heat shock protein family B member 1) can cause a group of length-dependent peripheral axonal neuropathy, including Charcot-Marie-Tooth disease type 2F (CMT2F) and distal hereditary motor neuropathy (dHMN). These two peripheral neuropathies are classified according to the presence or absence of sensory involvement. CMT2F has sensory symptom and (or) mild distal axonal sensory nerve degeneration in electrophysiology. In 2001, the pathogenic gene of CMT2F was identified on chromosome 7q11-q21 in a Russian family by linkage analysis [1]. In 2004, the missense mutations in *HSPB1* on chromosome 7 were first associated with CMT2F and dHMN [2]. To date, more than 30 mutations in the *HSPB1* have been found in patients with CMT2F and dHMN. Most of them are missense mutations, while a few are nonsense mutations or frameshift mutations [3]. The inheritance patterns caused by *HSPB1* mutation are mostly autosomal dominant (AD). In 2008, Houlden, H., et al. first reported CMT2F inherited in the way of autosomal recessive (AR) (p.L99M) [4]. In the following years, studies and reports supplemented this conclusion [5]. The clinical phenotypes caused by *HSPB1* gene mutation have certain heterogeneity in symptoms and onset age [6]. Even in the same family, clinical manifestations varied from muscle cramps as the only presenting symptom to a classic distal motor-sensory axonal neuropathy [7]. Peripheral axon degeneration was cardinal features in the established opinion. However, upper motor neuron involvement presenting with spastic paraplegia, hyperreflexia, and pyramidal signs have been continuously reported [3,8]. So far, Seven *HSPB1* mutations have been found in seven unrelated patients with sporadic amyotrophic lateral sclerosis (ALS); among them, bulbar symptoms were relatively common (5/7), and cognitive impairment was found in two patients [3,9,10,11]. The mutations in *HSPB1* constituted the most common cause of dHMN in various population with a frequency about 8~14.3% of clinical diagnostic dHMN [12,13,14]. As for axonal Charcot-Marie-Tooth disease, a cohort study from Italy showed that *HSPB1* mutation account for about 4% clinical identified CMT2 patients [13]. HSPB1, as an important chaperone protein, is also closely related to a variety of other diseases, including lung cancer, colorectal cancer, diabetes, ocular diseases, coronary heart disease and atrial fibrillation [15,16,17,18,19,20].

HSPB1, also known as HSP27 (the protein molecular weight is 27KD), located on 7q11.2, is a member of the sHSP family. It is constituted by a highly conserved α-crystallin domain (ACD) and the N-terminus and C-terminus in flank of ACD. The ACD, which has 90 amino acids, is characteristic of sHSP family, while N-terminus and C-terminus evolved independently and thereby is distinct from other sHSP [21]. HSPB1 is ubiquitously expressed, but predominantly in the nervous system, heart, and skeletal muscle [22,23]. As a molecular chaperone, HSPB1 plays many important roles in maintaining cell structure and functions. It can respond quickly when cells suffer from stress, exert anti-aggregation activity by combining with misfolded and easily aggregated substrates, and deal with substrates in cooperation with degradation system [22]. With these molecular properties, HSPB1 participates in the maintenance of the cytoskeleton [24], cell viability [25], neurofilament assembly [2], axonal transport [24,26], mitochondrial function [27], endoplasmic reticulum stress (ER stress) [17], autophagy [28], apoptosis [7,29], and so on.

In this study, we present three dHMN patients with *HSPB1* variations. c.C544G (p.P182A) and c.379C > T (p.R127W) have been reported in previous studies. c.G289C (p.V97L) is a novel variant and located in the α-crystallin domain. Through our study, we will provide new insights into the pathogenicity of the novel mutation and expand the clinical phenotypic spectrum and etiological spectrum of *HSPB1* gene mutation.

## 2. Methods and Materials

### 2.1. Clinical Data Collection

There are 3 hereditary peripheral neuropathy families from Shanghai Jiao Tong University Affiliated Sixth People’s Hospital in this study. Detailed clinical history collection and electrophysiological examination were carried in 3 probands. This study was approved by the hospital ethics committee and all participating members signed written informed consent.

### 2.2. Sural Nerve Biopsy and Ultrastructural Observation

Sural nerve biopsy was performed on proband 2 (at the age of 59) and the mother (at the age of 45) of proband 1. Nerve tissue was fixed in 2.5% glutaraldehyde solution with pH 7.2 and 4 °C overnight, fixed with 1% osmium tetroxide buffer for 2 h and embedded in Epoxy 618 resin. Toluidine blue staining was performed on semithin sections (0.7–1.0 μm) and observed with transmission microscopy (PHILIPS CM-120, Amsterdam, The Netherlands).

### 2.3. Genetic Analysis

The peripheral blood of probands and available family members in 3 families were collected for genetic analysis. A Qiagen kit (Hilden, Germany) was used to extract DNA from peripheral blood. Whole exome sequencing (WES) was used for found pathogenic mutation and carried in the Illumina HiSeq 2000 platform (San Diego, CA, USA). All results were confirmed with Sanger sequencing. The pathogenic of candidate variants was classified according to the American College of Medical Genetics and Genomics (ACMG) guidelines.

### 2.4. Cell Culture

SH-SY5Y cells, human neuroblastoma cell line, were obtained from American type culture collection (ATCC, Manassas, VI, USA) and cultivated in DMEM (Dulbecco’s Modified Eagle Medium, Gibco, Grand Island, NY, USA) with 10% fetal bovine serum (FBS) and 1% penicillin–streptomycin in a humidified incubator at 37 °C with 5% CO_2_, and the medium was changed every two days. After reaching 80% confluence, the cells were split with 0.25% trypsin and sub-cultured for further passages.

### 2.5. Transfection

Plasmid vector (PAcGFP-N1) carrying a full length of wild-type HSPB1 cDNA, was designed in our laboratory and constructed by manufacturer (Zorin, Shanghai, China). V97L mutation was induced by using site-directed mutagenesis kit (Agilent Technologies, Santa Clara, CA, USA) to obtain a mutant HSPB1 vector with the same framework. SH-SY5Y cells were cultured in 96-well plates overnight and transfected with plasmid vector carrying p.V97L HSPB1 mutation or wild-type HSPB1 cDNA by Lipofectamine 3000 reagent (Invitrogen, Waltham, MA, USA), according to the manufacturer’s instructions. Briefly, mutant p.V97L HSPB1 vector (0.1 µg) was diluted in 5 μL of reduced-serum medium Opti-MEM^®^ I Medium (Gibco, Waltham, MA, USA), mixed with an equal volume of Lipofectamine 3000 Reagent (5 μL) and incubated for 10–15 min at room temperature. The DNA–Lipofectamine complex was then added to SH-SY5Y cells and incubated for 24 or 48 h.

### 2.6. Cell Viability Assay and Morphological Analysis

Cell viability was measured with the CCK-8 assay (HY-K0301, MCE) using 96-well culture plates. Briefly, after transfection for 24 or 48 h and after being treated with H_2_O_2_ (300 µM) for 24 h, the cells were incubated with CCK-8 solution (10 µL) for 2 h at 37 °C and the absorbance at 450 nm was read using a microplate reader (Thermo Scientific, Waltham, MA, USA). The assays were performed three times in triplicate for each group. For morphological analysis, SH-SY5Y were imaged with the Olympus microscope system (Olympus, Tokyo, Japan) at 20× magnification.

### 2.7. TUNEL Assay

SH-SY5Y cells were transfected with DNA–Lipofectamine complex for 48 h and treated with H_2_O_2_ (300 µM) for 24 h. After treatment, the cells were fixed with 4% PFA for 30 min and washed with PBS 3 times followed by incubation in 0.3% Triton-X-100 PBS. After washing with PBS 3 times, SH-SY5Y were cultured with TUNEL solution (C1089, Beyotime, Shanghai, China) at 37 °C for 1 h in the dark, following the manufacturer’s instruction. All images were captured with a fluorescence microscope (Olympus, Tokyo, Japan).

### 2.8. Western Blot Analysis

After transfection and treatment, cultured cells were harvested and lysed in RIPA buffer supplemented with protease and phosphatase inhibitors on ice for 10 min and centrifuged at 12,000 rpm for 30 min at 4 °C. Supernatants were collected and the total protein concentration was evaluated using a BCA kit. Thereafter, an equal amount of protein (30 µg/well) was separated by sodium dodecyl sulphate-polyacrylamide gel electrophoresis (SDS-PAGE) and transferred to polyvinylidene fluoride (PVDF) membranes for 60 min at 250 mA. After membrane blocking with 5% non-fat milk, the membrane was incubated with primary antibodies (all from Cell Signaling Technology, Danvers, MA, USA; bax, 1:1000, #41162; bcl2, 1:200, #15071; caspase3, 1:1000, #9661) in TBS-T at 4 °C overnight followed by incubating with horseradish peroxidase (HRP)-conjugated immunoglobulin G (IgG) secondary antibodies (Beyotime; Goat anti rabbit IgG, 1:5000, A0208; Goat anti mouse IgG, 1:5000, A0216). β-tubulin primary antibody (Cell Signaling Technology, 1:1000, #2146) was used for normalization. The protein bands were detected using the ECL western blot detection system according to the manufacturer’s instruction. Digital images were analyzed using ImageJ gel analysis software to obtain the grey-scale value of signals.

## 3. Results

### 3.1. Clinical Data

Proband 1: F1-III-3 (Figure 1A) was a 14-year-old male with progressive bilateral lower limb weakness and walking difficulty from 10 years of age. He was born of full-term vaginal delivery with a normal developmental milestone. The initial weakness was illustrated by walking on flat ground with an abnormal posture and gait, and frequent tripping. He needed assistance in climbing stairs. There was a difficulty in standing up from squatting. He had a slowly progressive foot drop. There was no obvious limb muscle wasting or sensory disturbances. The patient’s academic performance was medium while his physical performance was significantly worse than peers. On examination, there was no obvious muscle atrophy. Contracture of Achilles tendons were observed in bilateral and more obviously in the left. He had reduced strength in distal lower limb (MRC 4/5), but normal strength in upper limb and proximal lower limb. Mildly decreased pinprick sensation below 10 cm of the left knee joint and 5 cm of the right knee joint was found. Reflexes were absent at the ankles and diminished at the knees and upper limb. Dysfunction of cranial nerve, pathological plantar reflex and ataxia were not found on examination. Mental psychological and cognitive tests were normal. His mother, F1-II-2, was 46 years old with a progressive walking weakness from age of 29. Her lower limb and hand had a significant muscle atrophy (Figure 1G,H). She sometimes complained of numbness in her fingers.

Nerve conduction studies (NCS) performed at age 13 showed reduced compound muscle action potential (CMAP) amplitude with normal motor nerve conductive velocity (MCV) in examined median, ulnar, tibial, and common peroneal nerve, the reductions of CMAP amplitude were more remarkable in the lower limb. The sensory nerve action potential (SNAP) amplitude and sensory nerve conduction velocity (SCV) were normal in all examined limbs. EMG showed spontaneous potential activities such as fibrillation wave and normal phase wave were observed in the distal of the limb. The MUP time limit was widened with unchanged amplitude, the polyphase wave was significantly increased. Some distal muscles showed simple phase in heavy contraction. Sural nerve biopsy of F1-II-2 (Figure 2A) displayed a mild to moderate decrease in large diameter myelinated nerve fibers, nerve fiber regeneration and clusters of regeneration. Increased mitochondrion gather (not shown here) was observed in some regenerated myelinated nerve fibers, thin myelinated fibers and atypical onion balls formed. According to clinical history, examination and electrophysiology, the diagnosis of dHMN was established.

WES was used for pathogenic gene screening. A heterozygous c.C544G (NM_001540) on exon 3 of *HSPB1* gene had been found in F1-III-3 (Figure 1B), resulting in the 182th amino acid of HSPB1 changing from proline to alanine (p.P182A). The variant had been reported previously [30]. There were also pathogenic reports about other variants forms in the same codon (p.P182L and p.P182S) [2,31]. His mother (F1-II-2) and elder female cousin (F1-III-1) carried the mutation, his father was wild type.

Proband 2: F2-II-7 (Figure 1C), a 65-year-old male, was investigated with complaints of weakness in his right lower limb and nocturnal leg cramps from the age of 51. There was a significant decrease in his weight in the initial 10 years (from 80 to 50 kg). The ability to walk and climb stairs were preserved. At the age of 57, he suffered a fracture of his left ankle and therefore accepted surgery. He felt weakness in his left leg 3 months later. After a year, the patient complained that his walking ability was only half of what it was originally. There were no sensory disturbances during his disease course. A sural nerve biopsy (Figure 2B) performed at the age 59 showed a decrease in myelinated nerve fibers, and some nerve fiber regeneration was observed in the nerve bundle, which is in accord with chronic peripheral nerve axonal damage. Diagnostic treatment was therefore given to the patient with prednisone 30 mg/d but there was no significant improvement. He consumed alcohol 100–200 g/d for 30 years and quit from the onset of leg weakness. His son (F2-III-3) developed nocturnal leg cramps at the age of 43. His brother (F2-II-3) and his nephew (F2-III-2) had progressive walking difficulty.

On examination, no obvious muscle wasting was found in limb. Foot drop was observed in the bilateral without significant foot deformity. He had normal strength in the upper limbs and proximal lower limb, but reduced strength in the distal lower limb (MRC 4/5). Reflexes were absent at the ankles and knees, and diminished at the upper limb. The sensory examination was normal. No obvious abnormality was found in the cranial nerve. Pathological plantar reflex and ataxia were not found on examination. Mental psychological and cognitive tests were normal.

NCS performed at the age of 59 showed significantly reduced CMAP amplitude in the bilateral common peroneal nerve and slightly reduced MCV in the right common peroneal nerve, and absent motor response in bilateral tibial nerve. The SNAP amplitude and SCV were normal in all limbs. EMG displayed fibrillation wave and positive sharp wave in the bilateral anterior tibial muscle and medial peroneal muscle, MUP showed prolonged duration with increased amplitude, polyphase wave increased, and recruitment potential reduced. According to the findings of the clinical history, examination and electrophysiology, the diagnosis of dHMN and restless legs syndrome (RLS) was established.

WES was used for pathogenic gene screening. A heterozygous missense mutation c.379C>T (p.R127W) in *HSPB1* gene were found in proband 2 (Figure 1D). The variant has been previously associated with CMT2 and dHMN [2,8]. No available blood samples from other family members were used for segregation validation.

Proband 3: F3-IV-1 (Figure 1E). This 50-year-old female developed gradual onset left lower limb weakness at the age of 44. Her right lower limb began to be affected 3 years later. She developed bilateral lower limb muscle atrophy in later years without any sensory symptoms. Her leg weakness gradually progressed to the point where she lost the ability to stand and walk. Her father, F3-III-2, had weakness in the bilateral lower limb. He was diagnosed as “lumbar disc herniation”. He accepted surgery and died shortly afterwards. Her grandfather also had a history of lower limb wasting.

On examination, she had severe wasting of the distal lower limb (Figure 1I). There was weakness of the proximal lower limb (MRC 2/5) and the distal lower limb (MRC 1/5) whilst the upper limb power was normal. Reflexes were absent at both ankles and knees and diminished at the upper limbs. The examination of cranial nerve and sensory examination were normal. No pyramidal signs and ataxia were found on examination.

NCS showed significantly reduced CMAP amplitude with slow MCV in the common peroneal nerve, and an absence of motor response in the bilateral tibial nerve. The SNAP amplitude and SCV were normal in all limbs. EMG revealed chronic neurogenic impairments with some fibrillation potentials and positive sharp wave in upper and lower limb muscles, T8 paraspinal muscle, sternocleidomastoid muscles, rectus abdominis. Based on the findings of the clinical history, examination and electrophysiology, the diagnosis of dHMN was established.

WES revealed a novel missense mutation in the *HSPB1* gene, a heterozygous change of c.G289C (p.V97L) on exon 1 of *HSPB1* gene (Figure 1D). The variant was not identified in dbSNP, 1000 Genome Project database and ExAC database. p.V97L was predicted to be possibly damaging by PolyPhen2 (with a score of 0.842, sensitivity: 0.83, specificity: 0.93), neutral by SIFT (SIFT score: −1.934), and disease caused by Mutationtaster. Her uncle (F3-III-8) and cousin (F3-IV-4) harbored the mutation, her daughter was wild type.

### 3.2. Effect of HSPB1-p.V97L Mutation on H_2_O_2_-Treated SH-SY5Y Cells

To study the pathogenicity of HSPB1-p. V97L mutation, we investigated the effect of this novel mutation on cell survival using CCK-8 cell viability assay. SH-SY5Y cells were transiently transfected with plasmid vector expressing green fluorescent protein (GFP)-tagged p. V97L mutant HSPB1 or wild-type for 24 h or 48 h and then incubated with H_2_O_2_ (300 μM) for 24 h. We found that transfected with mutated HSPB1 or wild-type HSPB1 alone had little toxic effect on SH-SY5Y cells. However, the expression of mutant HSBP1 lowered the cell viability remarkably, especially at 48 h, compared to wild-type HSPB1 in H_2_O_2_-treated cells (Figure 3A). Thus, cells transfected with mutated HSPB1 or wild-type HSPB1 for 48 h followed by H_2_O_2_ (300 µM) incubation for 24 h were used in subsequent experiments.

The inverted phase-contrast microscope was used to further evaluate the effects of mutated HSPB1 and wild-type HSPB1 on H_2_O_2_-induced morphological changes. As shown in Figure 3B, the cells in control group presented a normal cellular morphology, which was spindle or polygonal and each cell had a reticular connection with surrounding cells and disclosed numerous elongations. In contrast, after 24 h treatment with 300 μM H_2_O_2_, cell connections disappeared and cell shrinkage appeared in mutant HSBP1 and wild-type HSPB1 group. Moreover, the number of cells in mutant HSBP1 group were significantly lower than that in the wild-type HSPB1 group. The above observations indicated that p.V97L mutant HSPB1 could exert neurotoxicity on H_2_O_2_-treated SH-SY5Y cells.

### 3.3. Effects of HSPB1-p.V97L Mutation on H_2_O_2_-Induced Apoptosis in SH-SY5Y Cells

It has been reported that HSPB1 functions as anti-apoptotic molecule through inactivation of the caspase cascade [32]. TUNEL assay was performed to investigate whether mutant HSBP1 promoted apoptosis. As shown in Figure 4A, TUNEL stained cells were hardly observed in mutant HSBP1 and wild-type HSPB1 group. Nevertheless, after 24 h treatment with 300 μM H_2_O_2_, TUNEL stained cells were observed much more in the mutant HSBP1 group than in the wild-type HSPB1 group. The expression levels of apoptotic markers were further investigated to see if mutant HSPB1 reduced cell viability through apoptosis. After treating with H_2_O_2_ for 24 h, the expression levels of apoptosis-related protein including bcl2, bax, and caspase-3 in SH-SY5Y cells 48 h after transfection with HSPB1-p. V97L and wild-type HSPB1 were quantified using Western blot. Results showed that the expression of pro-apoptotic proteins bax and caspase-3 were dramatically upregulated, while the expression of anti-apoptotic protein bcl2 was significantly downregulated in SH-SY5Y cells transfected with mutant HSPB1, compared with those cells transfected with wild-type HSPB1, suggesting that HSPB1-p.V97L mutation induced the cell death via apoptosis (Figure 4B–E).

## 4. Discussion

In this study we present three distal axonal peripheral neuropathy families who carried heterozygous mutation in *HSPB1* gene. Many forms of dHMN have minor sensory abnormalities [14].Although the mother of case 1 and case 2 had abnormalities in the sural nerve biopsy, we tend to diagnose it as dHMN rather than CMT2. There is often an overlap with the axonal forms of Charcot–Marie–Tooth disease (CMT2). Mutations in many genes can be the cause of both CMT2 and dHMN, such as HSPB1, GARS, IGHMBP2, TRPV4, DNM2, DYNC1H1 [14]. Some patients with a dHMN can manifest minor sensory symptoms or signs that might appear after years of disease evolution. Electromyography and pathological examination help to find subclinical evidence of sensory nerve involvement in dHMN. However, it is generally believed that the sensory involvement of dHMN is not as obvious as that of CMT2. As shown in Figure 5, more than 40 pathogenic mutations of *HSPB1* have been reported. Most of these variants are related to CMT2F or dHMN. However, upper motor neuron (UMN) involvement presenting with spastic paraplegia, hyperreflexia, pyramidal signs, even ALS-like phenotype, were continuously reported. It is worth noting that patients with the same mutation can have different types of affected neurons. For example, patients carried the p.R127W of *HSPB1* can manifest as axonal peripheral neuropathy, but also ALS in some. Pathogenic mutations of three probands reported here are located on three exons of *HSPB1*. The clinical manifestations are not only dominated by lower limb weakness as a common feature, but also presented exclusive characteristics: case 1 had an early onset age, case 2 was accompanied with RLS, case 3 had multiple muscle groups involved in EMG.

The first proband had an early onset age and harbored the p.P182A mutation, which was located in the variable C-terminal domain and which has already been well descried in patients with dHMN or CMT2 [30,33]. The patient’s mother carried the same mutation, onset at the age of 29; this implied heterogeneity in the onset age, even in the same family. Other mutation forms (p.P182S and p.P182L) in the 182th codon also has been previously reported in patients with dHMN [2,31]. Previous reports showed that sensory involvement was rarely seen in patients with p.P182S or p.P182L but more frequently in p.P182A. However, patient 1 carried the p.P182A variant and had no sensory impairment in electrophysiology. Functional analysis showed reduced chaperone-like activity and altered oligomeric structure of HSPB1 in p.P182S and p.182L mutants [34,35].

The second proband with dHMN carried the p.R127W mutation, which located in the conserved ACD and has already been well described in patients with dHMN or CMT2 [2,8,36]. Another missense mutation at the same codon (p.R127L) has also been reported [37]. It is worth noting that proband 3 and his son developed nocturnal leg cramps, establishing a diagnosis of RLS. Nocturnal leg cramps had also been reported in a patient with late-onset dHMN II, who carried c.404C>G (p.S135F) in *HSPB1* [38]. Furthermore, cramps as a special symptom have been reported previously in Italian and Chinese Han patients with p.R127W mutation [8,39]. We speculate muscle cramps as a relatively common symptom in p.R127W mutant patients. However, the correlation between the genotype and clinical phenotype needs more data to be proved. A functional study of p.R127L mutation showed reduced survival of mutant fibroblasts when exposed to heat shock [37].

The third proband was diagnosed with dHMN; she carried *HSPB1* missense mutation (p.V97L), which has not been reported previously, and was located in the highly conserved ACD. The variant was not present in gnomAD, 1000 Genome Project database and ExAC database. It was predicted to be pathogenic using in silico tools including PolyPhen2 and Mutationtaster whilst neutral in SIFT. The proband 2 had an asymptomatic younger female cousin carrying the p.V97L mutation. One possible reason is that the cousin was under the age of onset; the other is incomplete penetrance of the mutation. Therefore, the follow-up study in this pedigree will be important. An adjacent variant (p.L99M) of p.V97L was reported in a consanguineous Pakistani family who was diagnosed with dHMN and presented AR inheritance [4]. Mechanism research of p.L99M mutant showed overproduction of larger and less stable homooligomers and reduced chaperone-like activity [40]. The functional analysis demonstrated that p.V97L mutation reduced cell viability under oxidative stress. But notably, cells transfected with mutant HSPB1 displayed no significance in cell viability compared to wild type under unstressed conditions. Therefore, we conjecture oxidative stress may constitute a trigger or promoting factor that influences disease onset and progress in patients carried pathogenic *HSPB1* mutation. The cascade reaction induced by oxidative stress has been well-described in other variants; Kalmar et al. observed decreased mitochondrial complex I activity, increased superoxide, impaired mitochondrial glutathione levels, and increased nitrotyrosine expression in S135F mutant mouse motor neurons in response to antimycin A treatment [26]. In addition to elevated reactive oxygen species (ROS) caused by extracellular drugs stimulation, wild type HSPB1 protein also can protect cells from ROS caused by huntingtin [27]. To further explore the mechanism underlying reduced cell viability in mutant HSPB1, the levels of apoptosis and apoptotic markers were investigated. As a result, cells transfected with mutant HSPB1 displayed no significant apoptosis and apoptotic markers compared to wild type under unstressed conditions, while increased apoptosis, upregulated pro-apoptotic proteins (bax and caspase-3), and downregulated anti-apoptotic protein (bcl2) were observed in cells transfected with HSPB1-p.V97L mutation under H_2_O_2_ stimulation, these findings were in accord with the results of cell viability assay and indicated that the mutation induced cell death via apoptosis. Elevated apoptosis has also been seen in other *HSPB1* mutations (p.T139M) [7]. Wild type HSPB1 can also exert anti-apoptotic functions in HeLa cells transfected with mutant HSPB5; these variants elevated cellular apoptosis level and were associated with congenital cataract and cardio-myopathy [41]. Similarly, overexpression of HSPB1 has a potent protective anti-apoptotic effect against the damaging effects from α-synuclein in cell models of Parkinson’s disease [42]. The possible mechanism underlying the anti-apoptotic effect of HSPB1 were as demonstrated by Bruey J.M. et al.: the cytochrome c released from the mitochondria to the cytosol was regarded as a hallmark of apoptosis, cytosol cytochrome-c mediated the interaction of Apaf-1 with procaspase-9 afterwards, and HSPB1 prevented the latter process by binding to cytosol cytochrome c [29]. Of note is that EMG revealed multi-muscle involvement in proband 3, which is very similar to ALS. However, due to insufficient evidence of UMN involvement, lower limbs-onset but not upper limbs, no respiratory muscle involvement, no bulbar symptoms, we therefore diagnosed the proband 3 as dHMN rather than ALS.

In sum, this study reported heterogeneous clinical phenotypes of dHMN caused by *HSPB1* mutation in three patients and describes a novel p.V97L mutation of HSPB1. Functional studies showed that reduced viability and increased apoptosis in cells carrying p.V97L mutation may contribute to pathogenesis.

## Figures and Tables

**Figure 1 biomolecules-12-01382-f001:**
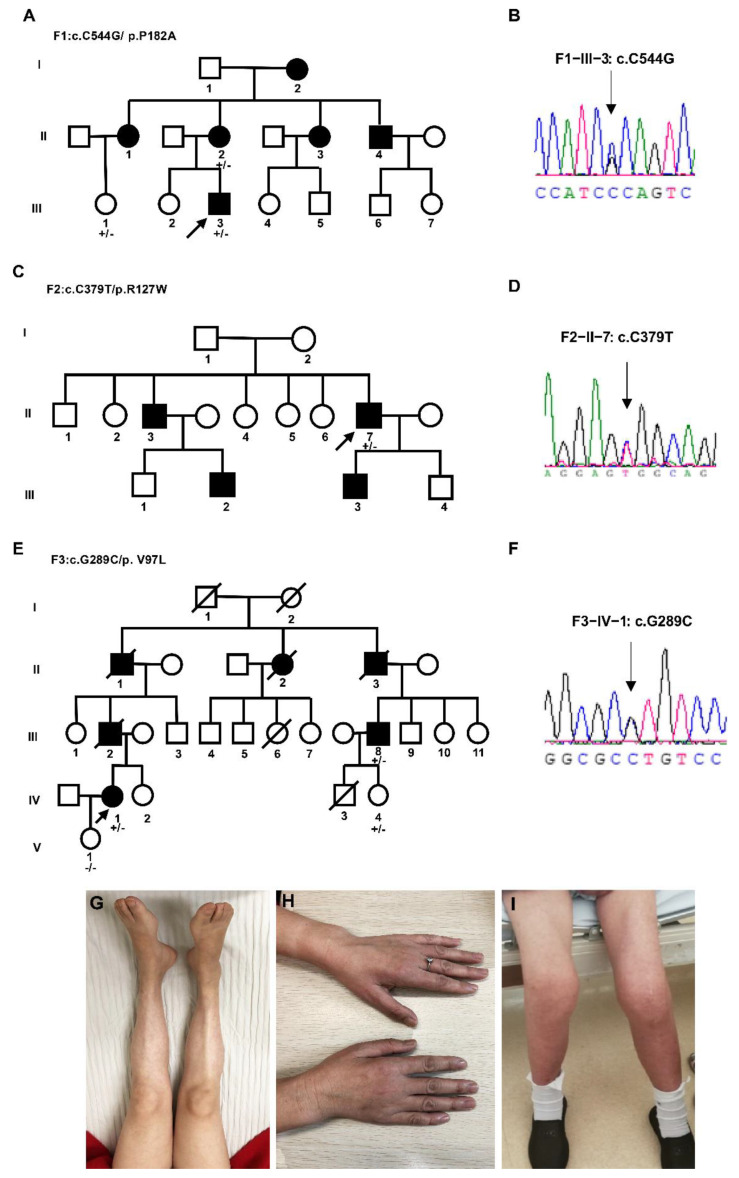
The family tree of three families (**A**,**C**,**E**), the Sanger sequencing verification results of probands (**B**,**D**,**F**), and clinical characterizations (**G**–**I**). (**G**) Tibialis anterior muscle and first dorsal interosseous muscle (**H**) atrophy of F1−II−2. (**I**). Bilateral proximal and distal lower limbs atrophy of proband 3. An arrow indicates the proband; square, male; circle, female; a filled symbol indicates affected; and an arrow through a symbol indicates the individual is deceased; +/− heterozygous, −/− homozygous normal for the *HSPB1* mutation.

**Figure 2 biomolecules-12-01382-f002:**
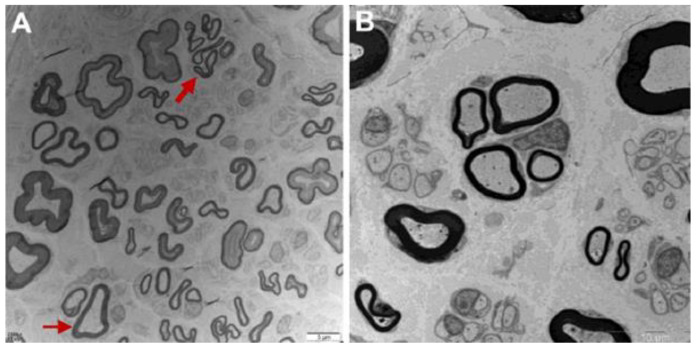
Sural nerve biopsy of F1-II-2 (**A**) and F2-II-7 (**B**). (**A**) Mild to moderate decrease of large myelinated fibers, clusters of regeneration (thick arrow) and demyelination (thin arrow) (bar = 5 μm). (**B**) Decrease of myelinated nerve fibers with some nerve fiber regeneration (bar = 10 μm).

**Figure 3 biomolecules-12-01382-f003:**
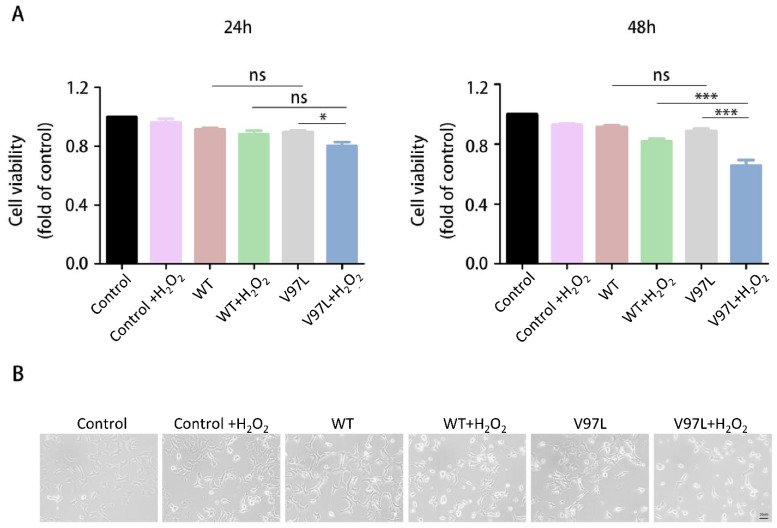
Effect of HSPB1-p.V97L mutation on H_2_O_2_-treated SH-SY5Y cells. (**A**) Effect of HSPB1-p.V97L mutation on cell survival using CCK-8 cell viability assay. SH-SY5Y cells were transiently transfected with plasmid vector expressing p.V97L mutant HSPB1 or wild-type HSPB1 for 24 h or 48 h and then incubated with H_2_O_2_ (300 μM) for 24 h; (**B**) cell morphology was observed with an inverted phase-contrast microscope. Control = SH-SY5Y cells. Data are presented as the means ± SEM from three independent experiments performed in triplicate. ns, no significance; * *p* < 0.05; *** *p* < 0.001. Scale bar = 50 µm.

**Figure 4 biomolecules-12-01382-f004:**
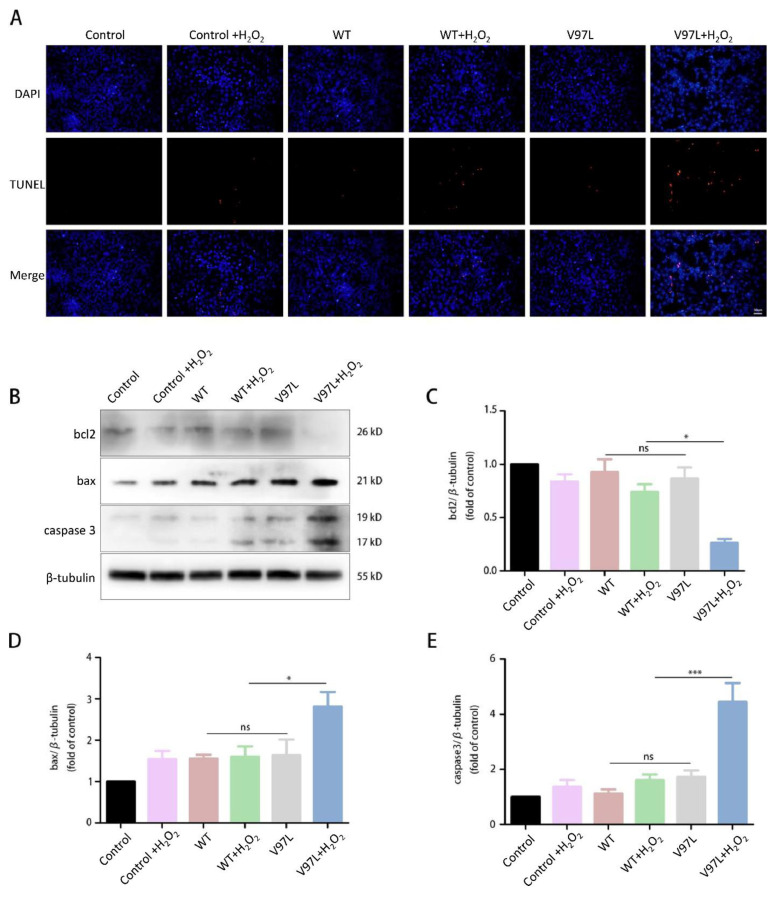
Effects of HSPB1-p.V97L mutation on H_2_O_2_-induced apoptosis in SH-SY5Y cells. Cells were transfected with mutated HSPB1 or wild-type HSPB1 for 48 h followed by H_2_O_2_ (300 µM) incubation for 24 h. (**A**) Representative photographs of fluorescent staining for DAPI (blue) and TUNEL (red). (**B**) Representative bands of western blot. (**C**–**E**) Quantitative analysis of the western blot bands. Control = SH-SY5Y cells. Data are presented as the means ± SEM of three independent experiments. ns, no significance, * *p* < 0.05; *** *p* < 0.001. Scale bar = 50 µm.

**Figure 5 biomolecules-12-01382-f005:**
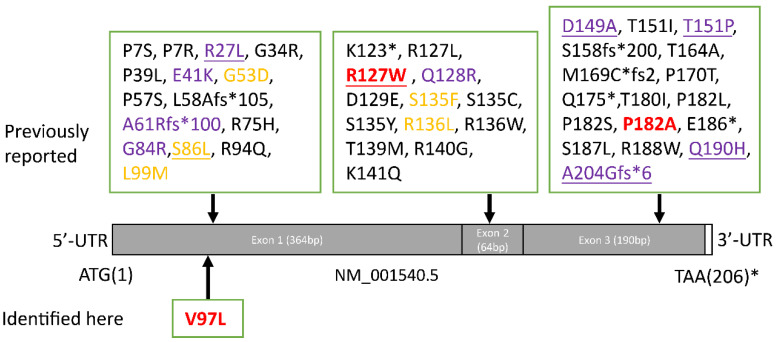
The schematic diagram of HSPB1 structure with mutations. Full length of HSPB1 (NM_001540.5) in brown consists of 205 amino acids. * stop codons; red, reported here; yellow, AR inheritance; purple, UMN involvement; underline, ALS-like phenotype.

## Data Availability

The data that support the findings of this study are available from the corresponding author upon reasonable request.

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
