# Peer review of "Heterogeneous Clinical Phenotypes of dHMN Caused by Mutation in HSPB1 Gene: A Case Series"

_biomolecules, 2022, doi:10.3390/biom12101382_

Round 1
Reviewer 1 Report
Well done and in-depth article both from a clinical and genetic point of view. The novel p.V97L mutation identified in proband 3 has been well analyzed, mainly to define its probable pathogenic effect.
The authors could submit the 3 mutations identified in the 3 families in one database, as LOVD (https://databases.lovd.nl/shared/genes). This would be useful especially for the novel mutation p.V97L in the HSPB1 gene identified in proband 3.
Figure 4 (A). would it be possible to lighten or improve the fluorescent staining for DAPI (blue) and TUNEL 276 (red)?
Figure 5. It is better to add in the legend also that the asterisk corresponds to the stop codon
Reviewer 2 Report
This study describes the clinical phenotype of 3 patients with an HSPB1 mutation, one p.V97L) of which is novel. To prove that this novel mutation is pathogenic, they employed cell assay studies that showed reduced viability and increased apoptosis in cells that carried this mutation.
The clinical details are exhausting and unnecessary.
In the results section:
With regard to F1-II-2:
1. She had normal sensory conduction studies. So why was a sural nerve biopsy performed?
2. In the biopsy description, what does “post-regeneration phenomenon” refer to?
3. ”Increased mitochondria… in some regenerated myelinated nerve fibers” is noted, but this is actually not seen in the figure.
4. Eventually, “the diagnosis of dHMN was established” but the changes seen in the sural nerve suggest a severe sensory component! How does that with the motor neuropathy diagnosis?
With regard to F2-II-7:
1. “There were no sensory disturbances during his disease course. Sural nerve biopsy (Figure 2B) performed at the age 59 showed”… What was a sensory nerve biopsy performed?
2. “monocyte infiltration” is noted but can’t be seen in the figure!
With regard to Figure 4. Effects of HSPB1-p. V97L mutation:
1. I expect to see the amount of HSPB1 on the blot as well.
2. Did it increase after transfection?
3. Was the level of the protein similar for the wild-type and the mutated?
The English is not good.
Reviewer 3 Report
Shen et al. submitted a case series of three patients classified as distal hereditary motor neuropathy (dHMN ). The aim of the paper was to demonstrate clinical heterogeneity of the patients and to describe a novel mutation in heat shock protein family B member 1 (HSPB1) gene. It is known that mutations in HSPB1 cause axonal neuropathies including Charcot-Marie-Tooth disease type 2F and dHMN . The strength of the paper is the demonstration of the pathogenicity of the novel mutation: the pathogenicity was confirmed by 2 out of 3 "in silico mutation analyses tools" and with functional studies: reduced cell viability assay of SH-SY5Y cells transfected with mutant HSPB1 compared to wild type under oxidative stress and with increased levels of apoptosis in mutant HSB1 cells under oxidative stress . Apoptosis was demonstrated by the TUNEL assay and by western blot. The latter showed unregulated levels of pro-apoptotic proteins (bax and caspase-3) and downregulated anti-apoptotic protein (bcl2).
The cascade reaction of elevated apoptosis triggered by oxidative stress and has been well described in the known mutations of HSPB1 before and fits with the proposed pathogenetic mechanism in this disease, which is nicely explained.
The clinical description and methods used are appropriate, however the authors should explain why they classified family 1 and 2 as dHMN despite demonstrating abnormal sural nerve, i.e. "abnormal pathomorhology of sensory nerve" in mother of proband 1 and in proband 2 and hypoesthesia in proband 1?
They should also comment if alcohol consumption in proband 2 could contribute to abnormal sural nerve biopsy?
Secondly, "increased apoptosis" demonstrated in Figure 4 with TUNEL assay would be more convincing if expressed as percentage of apoptotic nuclei as it seems that more nuclei are demonstrated by DAPI in V97L mutation+H202 than in WT+H202.
The manuscript is otherwise clear and presented in a well structured manner. The references are appropriate. Some minor changes in captions in Figure 3 and 4 should be made (in comments to the authors) to be more easily understandable. The conclusions are consistent with the arguments presented. Ethics statement is adequate and statement of data accessibility also.
Line 31: CMT2 and dHMN are for the first time mentioned in the main text and require explanation of the abbreviations, despite that they have been explained in the abstract
Line 83: add age of the patients when sural nerve biopsy was performed
Western-blot (Lines 124-136): blotting buffer, transfer conditions, suppliers of antibodies to be added
Line 136: normalisation to beta-tubulin should be mentioned
Line 147-148: Bold?
Line 157: please provide details: superficial and/or deep sensation affected
Line 163: was carpal tunnel syn. or similar in the mother excluded?
Line 181: explain why despite "sensory abnormalities" family1 is classified as dHMN
Line 198: could alcohol consumption in proband 2 explain sural nerve abnormalities? is abnormal sural nerve biopsy compatible with "pure" motor neuropathy
Figure 3: Please use instead of "Control" "Control (SH-SY5Ycells)" and instead "H2O2" = "Control (SH-SY5Ycells) +H202"
Figure 4: A caption - it seems from the qualitative assessment that more nuclei (DAPI positive) are in "mutants treated with H2O2" than in wild type treated with H202. Please express TUNEL positive nuclei as percentage
Discussion: discuss sensory abnormalities detected in some versus pure motor neuropathy
Round 2
Reviewer 2 Report
This study describes the clinical phenotype of 3 patients with an HSPB1 mutation, one p.V97L) of which is novel. To prove that this novel mutation is pathogenic, they employed cell assay studies that showed reduced viability and increased apoptosis in cells that carried this mutation.
The clinical details are exhausting and unnecessary.
Technical details were corrected in the current version, but the overall manuscript is long, and overly detailed leading.
the only new issue is the new mutation.